# Combining KPNA2 and FOXM1 Expression as Prognostic Markers and Therapeutic Targets in Hormone Receptor-Positive, HER2-Negative Breast Cancer

**DOI:** 10.3390/cancers17040671

**Published:** 2025-02-17

**Authors:** Tsen-Long Yang, Chung-Hsin Tsai, Ying-Wen Su, Yuan-Ching Chang, Fang Lee, To-Yu Huang, Fang-Yi Li, Po-Sheng Yang

**Affiliations:** 1Department of General Surgery, Shin Kong Wu Ho-Su Memorial Hospital, Taipei 111045, Taiwan; 2Department of General Surgery, MacKay Memorial Hospital, Taipei 104217, Taiwan; 3Department of Medical Oncology, MacKay Memorial Hospital, Taipei 104217, Taiwan; 4Department of Medicine, Mackay Medical College, Taipei 252005, Taiwan; 5Department of Medical Research, MacKay Memorial Hospital, Taipei 251404, Taiwan

**Keywords:** KPNA2, FOXM1, CCNB1, CCNB2, HR-positive, HER2-negative breast cancer

## Abstract

Breast cancer affects millions worldwide, with the hormone receptor-positive, HER2-negative (HR+HER2-) type being the most common form. Our study investigated the prognostic significance of FOXM1 expression specifically in HR+HER2- breast cancer patients with high KPNA2 levels. Through an analysis of clinical data from our hospital and public databases, we discovered that among patients with high KPNA2 expression, those with low FOXM1 levels showed significantly better survival outcomes. This novel finding suggests that FOXM1 expression could serve as a valuable prognostic marker specifically in KPNA2-high HR+HER2- breast cancer patients. While direct KPNA2 targeting remains challenging, our results indicate that FOXM1 inhibition could be particularly beneficial for patients with high KPNA2 expression. Several FOXM1 inhibitors, including thiostrepton and FDI-6, show promise as potential therapeutic strategies for this specific patient subgroup. This targeted approach based on both KPNA2 and FOXM1 expression patterns could lead to more personalized treatment strategies in HR+HER2- breast cancer.

## 1. Introduction

Breast cancer (BC) remains the leading malignancy among women globally [1], with over one million new cases annually and increasing projections [2]. In 2020, there were approximately 2.3 million cases of BC globally, resulting in 685,000 deaths. Projections suggest that by 2070, the number of cases will increase to 4.4 million [3]. Breast cancer classification is based on the gene expression patterns in molecular profiles, as established by Perou et al. in 2000 [4]. This study revealed the molecular heterogeneity of BCs through gene expression patterns, leading to the categorization of BC into four main groups: Luminal A (50–60% of cases), Luminal B (10% of cases), HER2-positive (20% of cases), and basal-like triple-negative tumors (approximately 10% of cases). In 2011, the St. Gallen expert consensus panel adopted a molecular subtype approach for BC treatment, utilizing immunohistochemistry (IHC) markers to identify intrinsic molecular subtypes and guide therapy [5]. Currently, semiquantitative IHC expression of estrogen receptor (ER), progesterone receptor (PR), Human Epidermal Growth Factor Receptor 2 (HER2), and proliferation index Ki-67 is used to define surrogates of the four molecular subtypes, with Luminal B tumors distinguished by higher histologic grade, lower PR expression (20% or less), and higher Ki-67 expression (typically ≥ 14%) [6,7]. However, IHC surrogates may not always accurately reflect the true intrinsic molecular subtype, with discordance rates between IHC analysis and genomic expression profiles reaching up to 30% [8]. Notably, approximately 20% of Luminal B cancers may be HER2-positive according to IHC analysis, adding to the complexity of breast cancer classification [9]. According to the latest AJCC Cancer Staging Manual, Luminal A and B tumors are identified as ER-positive, PR-positive, and HER2-negative by IHC, with all HER2-positive tumors classified as the “HER2 subtype” regardless of HR status. ER-positive, PR-positive, and HER2-positive tumors represent the “Luminal B HER2-positive” subtype.

Karyopherin alpha 2 (KPNA2), a nuclear transport protein, plays a crucial role in nucleocytoplasmic communication and the transport of tumor-related proteins, with its elevated expression strongly correlating with poor prognosis in breast cancer, particularly in aggressive subtypes [10,11]. FOXM1, an oncogenic transcription factor in breast cancer, regulates critical cell cycle transitions and DNA repair processes [12,13]. FOXM1 has been demonstrated to bind and regulate multiple target genes, including CCNB1, CCNB2, PLK1, ECT2, EZH2, GPSM2, KIF18B, KPNA2, PHF19, PSRC1, and STK17B, predominantly influencing the G2/M transition through direct transcriptional activation [13]. In breast cancer, studies show that FOXM1 directly regulates KPNA2 as a target gene, while KPNA2 reciprocally facilitates FOXM1’s nuclear transport, creating a regulatory loop that influences tumor progression [12]. The KPNA2 functions as an adaptor protein regulating FOXM1’s subcellular localization and activity [10], where high KPNA2 expression can lead to aberrant nuclear accumulation of FOXM1, activating the transcription of CCNB1/CCNB2, which is essential for G2/M phase transition [13]. The FOXM1 directly binds to and regulates CCNB1/CCNB2 promoters for breast cancer cell cycle progression [14], while KPNA2 modulates this process through its effects on FOXM1 nuclear transport and retention [15]. This intricate interplay between KPNA2 and the FOXM1-CCNB1/B2 axis not only provides insights into breast cancer cell cycle regulation but also suggests potential therapeutic targets in breast cancer treatment. In breast cancer, particularly in triple-negative breast cancer (TNBC), FOXM1 contributes significantly to treatment resistance by regulating DNA repair genes and protecting cancer cells from DNA damage, while the presence of both high FOXM1 and KPNA2 expression correlates with poorer clinical outcomes and increased resistance to various therapies, including endocrine treatments and chemotherapy [16,17,18,19].

The primary aim of this study was to identify effective prognostic biomarkers for breast cancer (BC), with a specific focus on hormone receptor-positive, HER2-negative (HR+/HER2-) breast cancer, which represents a major subset of BC cases. We investigated KPNA2 and FOXM1 as potential prognostic indicators and therapeutic targets. Our comprehensive analysis incorporated mRNA expression profiles from three distinct sources: MacKay Memorial Hospital (MMH) patient samples, The Cancer Genome Atlas (TCGA), and Gene Expression Omnibus (GEO) databases. The study examined the interrelationships between KPNA2, FOXM1, CCNB1, and CCNB2 expression levels, and evaluated their associations with established clinical markers including estrogen receptor (ER), progesterone receptor (PR), HER2 status, and Ki67 proliferation index, to determine their impact on patient outcomes.

## 2. Materials and Methods

### 2.1. Study Population and Sample Collection

Breast tumor specimens were collected from 184 patients with primary invasive BC (stage I–IV) (From August 2021 to July 2023). All patients were admitted to the Department of General Surgery, MacKay Memorial Hospital (MMH), Taipei, Taiwan, and were asked to submit written informed consent according to the study approved by the MMH Institutional Review Board (IRB approval number: 21MMHIS069e). After surgery, pathological material from different areas of the tumor was processed for conventional histological procedures, and the fresh samples were promptly preserved in RNAlater upon resection and stored following the manufacturer’s instructions. Clinical data, including ER, PR, HER2, Ki67 proliferation index, molecular subtype, pathology type, tumor histological grade and stage, were collected for all cases. Pathology of the type of tumors was carried out according to the criteria of the World Health Organization. Tumor stage was determined according to the guidelines of the AJCC ver. 8. Tumors were graded according to Bloom and Richardson’s modification of the work of Elston and Ellis [20]. Molecular subtypes were divided into Luminal A, Luminal B HER2-, Luminal B HER2 +, HER2, and triple negative (TNBC) [21]. Luminal A subtype was defined as being ER-positive, HER2-negative, and Ki67-low (<14% cells positive), and the Luminal B HER2- subtype as being ER-positive, HER2-negative, and Ki67-high (≥14% cells positive) [22].

### 2.2. RNA Isolation

Total RNA was extracted using Trizol^®^ Reagent (Invitrogen, Carlsbad, CA, USA) according to the instruction manual. Purified RNA was quantified at OD260 nm using an ND-1000 spectrophotometer (Nanodrop Technology, Wilmington, DE, USA) and qualified by using a Bioanalyzer 2100 (Agilent Technology, Santa Clara, CA, USA) with RNA 6000 LabChip kit (Agilent Technology, Santa Clara, CA, USA).

### 2.3. Library Preparation and Sequencing and Sequence Quality Trimming

All RNA sample preparation procedures were carried out according to the Illumina’s official protocol. SureSelect XT HS2 mRNA Library Preparation kit (Agilent, Santa Clara, CA, USA) was used for library construction followed by AMPure XP beads (Beckman Coulter, Brea, CA, USA) size selection. The sequence was determined using Illumina’s sequencing-by-synthesis (SBS) technology (Illumina, San Diego, CA, USA). Sequencing data (FASTQ reads) were generated using Welgene Biotech’s pipeline based on Illumina’s base-calling program bcl2fastq v2.20. Quality trimming was performed to remove low quality reads/bases. Lower quality bases from the 3′ end were removed using a sliding-window approach as the per-base quality gradually drops toward the 3′ end of reads. Both adaptor clipping and sequence quality trimming were performed using Trimmomatic v0.36 [23] with a sliding-window approach.

### 2.4. RNA Alignment Tool (HISAT2) and Differential Expression

HISAT2 is a fast and sensitive alignment program for mapping next-generation sequencing reads to genomes. The new indexing scheme of HISAT2 is based on the hierarchical graph FM index (GFM index). HISAT2 uses global GFM index and a large set of small GFM indexes that collectively cover the whole genome for rapid and accurate alignment. Furthermore, small GFM index sets show a great performance in splicing junction alignment, so HISAT2 could be an effective tool for transcriptome alignment. In the gene expression diagram, The genes were normalized to Transcripts Per Million (TPM) [24]. In the table of Differential Expression Analysis (DE table), Differential Expression Analysis was performed using StringTie (StringTie v2.1.4) and DEseq (DEseq v1.39.0) or DEseq2 (DEseq2 v1.28.1) [25] with genome bias detection/correction and Welgene Biotech’s in-house pipeline.

### 2.5. Tissue Microarray

The breast cancer tissue microarray was purchased from US Biomax (BR803c, Rockville, MD, USA). The tissue microarray was interpreted and scored according to the intensity of staining by pathologists. The scores were finally divided into three groups, low, middle, and high, based on statistical requirements.

### 2.6. Immunohistochemistry

Immunohistochemistry was carried out as previously reported [26] using an antibody against KPNA2 (GTX106323, Genetex, San Antonio, TX, USA).

### 2.7. Public Database

The expression of specific genes and survival information obtained from The Cancer Genome Atlas (TCGA) for Breast Cancer (BRCA) and for GSE7390 obtained from the Gene Expression Omnibus (GEO) database were used for the Kaplan–Meier survival analysis and gene set Enrichment Analysis (GSEA) with the medium levels of these genes used as cut-off points [27]. The gene expressions obtained from TCGA BRCA and GSE7390 were used for correlation analysis. The sources of gene expression profiling data of breast cancer and the corresponding clinicopathological parameters were downloaded from USCS Xena and GEO database. The sources of these gene expression profiling datasets are listed in Appendix A.

### 2.8. Statistical Analysis

Statistical analyses were performed using GraphPad Prism software version 10.1 (GraphPad Software, La Jolla, CA, USA). The patients were divided into two sub cohorts according to the cohort median of gene expression transcript levels. Descriptive statistics of clinical and pathological variables were reported using the frequencies for categorical variables or medians with minimum/maximum frequencies for continuous variables. Group-wise comparisons of categorical data were evaluated using the chi-squared test with two-tailed *p*-values generated using Fisher’s exact test. Unpaired *t*-tests and Mann–Whitney U-tests were used to analyze the intergroup differences. A *p*-value of <0.05 was considered significant. For survival analysis, Kaplan–Meier survival curves for overall survival (OS) and disease-specific survival (DSS) were plotted to compare the differences between the high and low gene expression groups using the log-rank test with reported *p*-values. The *p*-values reported in each Kaplan–Meier (K-M) plot were generated by conducting the log-rank (Mantel–Cox) test for the comparison of multiple survival curves to assess whether all survival curves for each subgroup are identical in the overall populations from which subjects in each group were sampled. If the *p*-values were small enough to reject the null hypothesis, it suggested that stratification by FOXM1 and another gene transcript levels was capable of differential outcome prognosis. A survival K-M analysis in the Prizm software (v. 10.1) (https://www.graphpad.com/guides/prism/latest/user-guide/survival_table.htm, accessed on 25 December 2024) also provided log-rank results for trend and the Gehan–Breslow–Wilcoxon test with respective *p*-values was used for the assessment of curve separation. We did follow up significant *p*-values with multiple pair-wise comparisons of subgroups (curves) or even Cox regression modeling to gain further and quantitative insights into biomarker potential.

Enrichment Analysis (GSEA) was conducted using the Broad Institute platform [24,25]. The ranked results of the GSEA indicated that gene expression that was positively correlated with KPNA2 in the primary tumors of the MMH cohort was highly enriched with the priori-defined “Hallmark_G2M_CHECKPOINT” gene set in the Molecular Signatures Database (MSigDB). The magnitude of overrepresentation is reflected by the maximal enrichment score (ES) based on a weighted running sum (green curve) and its significance by the empirical *p*-value based on 1000 permutations of the phenotype class labels. Patient samples were sorted based on the KPNA2 transcript levels to give two phenotype classes of “KPNA2 high” and “KPNA2 low”, comprising samples with KPNA2 expression exceeding the cohort median, and those with expression levels below the median, respectively. The default weighted Signal2Noise setting was applied to assign ranking metric scores (gray bar plot) based on the degree of correlation with phenotype rankings. The ES was subsequently normalized (NES) to account for the size of the gene set among all others in the MSigDB, and a false discovery rate (FDR) was determined based on the NES to control for false positive findings arising from multiple hypothesis testing.

## 3. Results

### 3.1. Patient Characteristics

A total of 184 patients with primary BC (From August 2021 to July 2023) were enrolled in this study. The age of patients at diagnosis varied from 26 to 92 years, with an average age of 55 years. Table 1 and Table 2 outline the clinicopathological data of the cases. Invasive ductal carcinoma was predominant, constituting 65% (120 out of 184 cases) of the study cohort. Approximately 65% (119 out of 184) of patients presented with Grade II tumors. Most patients (60%, or 110 out of 184) were diagnosed with stage I breast cancer, followed by 27% (50 out of 184) with stage II, and 13% (24 out of 184) with stage III/IV. Tumor size distribution revealed 45% (82 out of 184) as T1, 41% (76 out of 184) as T2, and only 3% (6 out of 184) as T3 or higher. Lymph node involvement (N1 and above) was observed in 27% (49 out of 184) of patients. An initial diagnosis of distant metastasis (M1, stage IV) accounted for 6% (11 out of 184) of cases. Hormone receptor positivity was notable in 78% (144 out of 184) of the ER-positive and 68% (126 out of 184) of the PR-positive cases. HER2 positivity was observed in 2.2% (4 out of 184) of cases, while the Ki67 proliferation index exceeded 14% in approximately 46% (85 out of 184) of cases. The intrinsic subtypes identified in the series were as follows: Luminal A (45%, or 82 out of 184), Luminal B HER2-negative (22%, or 41 out of 184), Luminal B HER2-positive (12%, or 22 out of 184), HER2-enriched (9%, or 16 out of 184), and TNBC (11%, or 20 out of 184).

### 3.2. KPNA2 Expression Is Positively Correlated with the Expression of FOXM1, CCNB1, and CCNB2

FOXM1 is a transcriptional regulator involved in many cellular reactions including cell differentiation and proliferation. The overexpression of FOXM1 in many cancers including breast cancer is associated with advanced tumor stage, higher proliferation, and poor prognosis [28,29]. Studies have demonstrated that FOXM1 regulates KPNA2 expression and the protein–protein interaction network between KPNA2 and FOXM1 [13,30]. We were then interested in exploring the role of FOXM1 in the development and progression of BC. The correlation analysis demonstrates that there was a strong positive correlation between the expression of FOXM1 and KPNA2 in the MMH database (r = 0.54 and *p* < 0.0001; Figure 1a, leftmost panel). Consistent with the MMH data, a positive correlation between FOXM1 and KPNA2 expression was observed in the TCGA cohort (r = 0.76 and *p* < 0.001; Figure 1b, leftmost panel) and GSE7390 dataset (KPNA2 probe: 211762_s_at, r = 0.70, *p* < 0.001; KPNA2 probe: 201088_at, r = 0.70, *p* < 0.001, Figure 1c, leftmost). KPNA2 expression is highly related to the FOXM1 signaling pathway in BC.

FOXM1 is involved in cellular activities through the activation of many target genes including CCNB1 and CCNB2. To further explore the FOXM1 signaling, we studied the expression of the KPNA2 and FOXM1 downstream genes CCNB1 and CCNB2. In accordance with the results obtained from the MMH cohort (Figure 1a, second and third from the left), the mRNA expression levels of CCNB1 and CCNB2 were significantly and positively correlated with KPNA2 expression in the TCGA database (Figure 1b, second and third from the left) and GSE7390 dataset (Probe: 211762_s_at and Probe: 201088_at, Figure 1c, second and third from the left). Collectively, the expression of FOXM1 and its downstream genes CCNB1 and CCNB2 is highly correlated with the expression of KPNA2 in BC.

### 3.3. High KPNA2 Expression Is Positively Correlated with the G2/M Checkpoint Pathway in BC Patient

To explore the functional pathway (GSEA) and expression correlation of KPNA2 and its associated genes (FOXM1, CCNB1and CCNB2) with BC cell growth, we analyzed the RNA-seq data of BC patients from our series (Figure 1d), from the TCGA dataset (Figure 1e), and the GSE7390 dataset (Figure 1f, KPNA2 Probe: 201088_at and 211762_s_at), and then divided the patients into two groups, high and low KPNA2 expression. From the GSEA results, we observed that these genes were highly related to the ‘HALLMARK_G2M_CHECKPOINT’ gene set, showing that G2/M checkpoint pathway was found significantly enriched in BC patients with KPNA2 high expression group. The results of the GSEA further indicate a significant correlation with KPNA2 pathway in the enrichment of the Molecular Signatures Database (MSigDB) Collection. The functional pathway (GSEA) and expression correlation analysis suggested that KPNA2 is functionally co-operative and transcriptionally co-regulated by FOXM1, CCNB1, and CCNB2 during the G2M checkpoint of cell cycle progression in our series (Figure 1d) and validated in the TCGA dataset (Figure 1e) and GSE7390 dataset (Figure 1f, KPNA2 Probe: 201088_at and 211762_s_at).

### 3.4. High KPNA2 Expression Is Correlated with Advanced Tumor Grade and Staging in BC Patients

A higher tumor grade indicates poorer differentiation and a faster growth and spread of cancer cells. From Table 1 and Appendix A, the high expression of KPNA2 was found to be correlated with tumor grade (*p* < 0.0001) in our series. This result was validated using the GSE7390 dataset, demonstrating a positive correlation between KPNA2 expression and higher BC tumor grades (Appendix A, Probes: 201088_at, *p* < 0.001 and 211762_s_at, *p* < 0.001). According to the AJCC system, high KPNA2 expression was correlated with stage (*p* = 0.0107). Validation of this finding showed that high KPNA2 expression is significantly associated with more advanced tumor stages in the TCGA-BRAC cohort (*p* = 0.004, Appendix A).

To further investigate the association between KPNA2 protein expression and tumor grade in BC patients, immunohistochemistry was performed on a BC microarray using an anti-KPNA2 antibody. Pathologists scored tissue microarrays based on staining intensity, categorizing scores into low, moderate, and high groups. Analysis revealed a significant association between KPNA2 protein expression and tumor grade in BC patients (*p* = 0.049, Appendix A).

### 3.5. Correlation of KPNA2, FOXM1, CCNB1, and CCNB2 Transcript Levels and the Pathological Assessment of ER-Positive or PR-Positive or Ki67-Positive Fractions and the HER2 Transcript Levels and Molecular Subtype of BC Patients

The molecular subtype of BC is based on immunohistochemical indicators such as ER, PR, HER-2, and the cell proliferation index Ki67 [21]. We tried to explore the correlation of KPNA2, FOXM1, CCNB1, and CCNB2 transcript levels and the pathological assessment of ER-positive or PR-positive or Ki67-positive fractions and the HER2 transcript levels in our series. Corresponding Pearson’s correlation efficiency and *p* values are presented in Figure 2 and Table 1 and Appendix A. In Figure 2, the KPNA2, FOXM1, CCNB1, and CCNB2 transcript levels are correlated with the pathological assessment of ER-positive or PR-positive (except CCNB1 and CCNB2) or Ki67-positive fractions and the HER2 transcript levels (except FOXM1) in our series. Further analysis of the MMH cohort revealed that a high or low expression (transcript levels lower or higher than the cohort median) of KPNA2/FOXM1/CCNB1/CCNB2 is associated with ER, PR, HER2, and Ki67 status, molecular subtype, tumor grade, and AJCC stage (Table 2 and Appendix A). Additionally, only FOXM1 was associated with pathology type. In the TCGA dataset, a high or low expression of these genes is associated with ER, HER2 status, T stage, and AJCC stage, and only FOXM1 was associated with the intrinsic subtype (Appendix A). Furthermore, we checked the relationships between KPNA2, FOXM1, CCNB1, and CCNB2 transcript levels and the molecular type of BC in our patients. The distribution of these four genes transcript levels among the molecular subtypes of our 184 BC patients are shown in Table 2 and Appendix A. The KPNA2 transcript levels were lower in Luminal A BC patients and significantly compared with Luminal B HER2 negative (*p*< 0.0001), with Luminal B HER2 positive (*p*< 0.0001), with HER-2 (*p*< 0.001), and with TNBC (*p* < 0.01) patients (Appendix A). These results were also found in FOXM1 (Appendix A), CCNB1 (Appendix A), and CCNB2 (Appendix A).

### 3.6. High KPNA2, FOXM1, CCNB1, and CCNB2 Expression Were Correlated with Poor Survival of BC Patients

To evaluate whether KPNA2 expression levels have predictive significance for BC prognosis, we analyzed the expression of KPNA2 mRNA and its relationship with the clinical outcomes. The prognostic potential of KPNA2 was assessed using RNA-seq data from TCGA and the GEO database. The expression level of KPNA2 was separated into high or low groups, and subgroup analyses were conducted based on the clinical variables including OS, disease-specific survival (DSS) and metastasis-free survival (MFS) (Figure 3a). As indicated in Figure 3a (upper left), a higher KPNA2 mRNA expression is significantly associated with a shorter OS in the TCGA cohort (*p* = 0.021). This result was validated with a GSE7390 dataset from the GEO database using two different probe sets measuring distinctive regions of the KPNA2 gene (Figure 3a, upper middle, probe: 211762_s_at, *p* = 0.032 and Figure 3a, upper right, probe: 201088_at, *p* = 0.002). BC patients with increased KPNA2 expression had a shorter survival rate in all cohorts tested.

In addition to OS, we also noticed a high expression of KPNA2 mRNA which was significantly related to poor DSS in the TCGA cohort (*p* = 0.009, Figure 3a, lower left). Since MFS is also a predictor of survival in various cancers, the association of KPNA2 expression and MFS was studied using the GSE7390 dataset with two different KPNA2 probes as mentioned above. Our results demonstrate that there is a significant correlation between high KPNA2 expression and poor MFS, as measured with the KPNA2 probe: 201088_at (*p* = 0.007, Figure 3a, lower right). In contrast, there seems to be no apparent association between KPNA2 expression and MFS using the other KPNA2 probe: 211762_s_at (*p* = 0052, Figure 3a, lower middle).

The individual associations of FOXM1, CCNB1, and CCNB2 with OS and DSS in the TCGA dataset were similarly assessed by the K-M plots (Figure 3b–d). Elevated levels of FOXM1 mRNA were notably linked to poorer OS and DSS within the TCGA cohort (Figure 3b, left), mirroring findings seen with the CCNB1 (Figure 3c, middle) and CCNB2 (Figure 3d, right) genes.

### 3.7. Combining KPNA2 and FOXM1 mRNA Levels Were Correlated with Poor Survival of BC Patient

To investigate the association between the combined expression levels of KPNA2, FOXM1, CCNB1, and CCNB2 and survival outcomes in BC patients, we analyzed data from the TCGA dataset using Kaplan–Meier plots. In the combined analysis of KPNA2 and FOXM1 expression, a total of 1094 patients were stratified into four groups: high FOXM1 and high KPNA2 (n = 449), high FOXM1 and low KPNA2 (n = 98), low FOXM1 and high KPNA2 (n = 99), and low FOXM1 and low KPNA2 (n = 448). Elevated levels of both KPNA2 and FOXM1 mRNA were significantly associated with poorer OS (Figure 4a, *p* = 0.0046) and DSS (Figure 4b, *p* = 0.006). When examining the combined expression of FOXM1 with either CCNB1 or CCNB2, we found that only a poorer DSS was observed. In the combining FOXM1 and CCNB1 group, a poorer DSS was noted (Figure 4d, *p* = 0.029), while OS was not significantly affected (Figure 4c, *p* = 0.1539). Similarly, in the combined FOXM1 and CCNB2 group, a poorer DSS was observed (Figure 4f, *p* = 0.0067), with no significant impact on OS (Figure 4e, *p* = 0.121).

### 3.8. Combined KPNA2 and FOXM1 mRNA Levels Were Correlated with Poor Survival of HR+Her-2- BC Patient

To delve deeper into the involvement of KPNA2, FOXM1, CCNB1, and CCNB2 expressions in BC molecular subtypes, we conducted an analysis of these four genes’ expression levels and their association with OS and DSS in the TCGA dataset, stratified by ER, PR, and HER2 status using Kaplan–Meier plots. We observed poorer DSS in patients with higher expression levels of KPNA2, FOXM1, and CCNB1 among Her-2 negative BC patients (Figure 5a), but not in the other ER, PR, and HER2 status groups (Appendix A). Specifically, Figure 5a (leftmost) illustrates a significant association between elevated KPNA2 mRNA expression and shorter DSS in the TCGA cohort (*p* = 0.0232). Similar associations were observed with high FOXM1 (*p* = 0.0082) and CCNB1 (*p* = 0.0477) expression groups, but not with high CCNB2 expression (*p* = 0.0505) (Figure 5a, rightmost picture). Regarding OS, the impact of high expression levels of these four genes was not significant and was consistent across all different ER, PR, and HER2 status subgroups of breast cancer patients (Appendix A). However, a lower OS was only observed in the higher KPNA2 mRNA expression group of PR+ BC patients, albeit with mild significance (the leftmost picture in the third row of Appendix A, *p* = 0.0496).

Moreover, in the analysis of combining KPNA2 and FOXM1 expressions among HR+HER2- BC patients, we categorized 555 individuals into four groups: high FOXM1 and high KPNA2 (n = 208), high FOXM1 and low KPNA2 (n = 49), low FOXM1 and high KPNA2 (n = 49), and low FOXM1 and low KPNA2 (n = 246). Interestingly, we discovered that the combination of high KPNA2 and low FOXM1 mRNA levels exhibited the most favorable outcomes in terms of overall survival (OS) (*p* = 0.0355) and disease-specific survival (DSS) (*p* = 0.0424) (Figure 5b). We further compared the OS and DSS of HR+HER2- BC patients between high KPNA2 and low FOXM1 expressions with the other groups; the high KPNA2 and low FOXM1 BC patients had significantly better OS and DSS rates (Figure 5b, second row). However, such associations were not observed in the combination of FOXM1 and CCNB1 group (Figure 5b, third row) or in the FOXM1 and CCNB2 group (Figure 5b, bottom row).

## 4. Discussion

Our study revealed strong positive correlations between KPNA2 and FOXM1, CCNB1, and CCNB2 expression, with a significant association with the G2/M checkpoint pathway in breast cancer patients. These genes correlated positively with ER status, PR status, Ki67 index, and HER2 levels across breast cancer subtypes. TCGA and GEO analyses showed that a high expression of these genes, particularly KPNA2 and FOXM1, predicted poor overall and disease-specific survival (*p* = 0.0046 and *p* = 0.006, respectively). Notably, HR+/HER2- breast cancer patients with high KPNA2 but low FOXM1 mRNA levels showed significantly better survival outcomes (OS: *p* = 0.0355; DSS: *p* = 0.0424), suggesting a novel prognostic signature for this major breast cancer subtype.

We further compared hormone receptor expressions between breast cancer subtypes Luminal A and B for 123 cases (82 Luminal A and 41 Luminal B) in this study. Statistical analysis revealed that Luminal A tumors exhibited a significantly higher expression of both estrogen receptor (ER) (mean: 92.2% vs. 81.6%, *p* = 0.0333) and progesterone receptor (PR) (mean: 63.4% vs. 46.9%, *p* = 0.0346) compared to Luminal B tumors. While both subtypes showed high median ER expression (95%), Luminal A demonstrated notably higher PR median values (80% vs. 50%). Interestingly, Luminal B tumors showed greater heterogeneity in ER expression, as evidenced by higher variance (744.87 vs. 106.06), suggesting more diverse ER expression patterns within this subtype despite its overall lower expression levels. In Figure 2 and Appendix A, we showed that the transcript levels of KPNA2, FOXM1, and/or CCNB1, CCNB2 were negatively associated with the ER or PR transcripts in the MMH cohort. Furthermore, in Appendix A, we also showed that these genes’ transcript levels were significantly lower in the Luminal A subtype compared to the Luminal B subtype. The negative correlation between KPNA2, FOXM1, CCNB1, and CCNB2 transcript levels and ER/PR expression aligns well with our findings that Luminal A tumors exhibit significantly higher ER and PR expression compared to Luminal B tumors. The observed pattern creates a coherent biological relationship where Luminal A tumors, characterized by higher hormone receptor expression, show lower levels of KPNA2, FOXM1, CCNB1, and CCNB2, while Luminal B tumors, with their relatively lower hormone receptor expression, demonstrate higher levels of these genes. This inverse relationship suggests these genes may play crucial roles in the molecular mechanisms distinguishing Luminal A from Luminal B breast cancers, potentially through their interaction with hormone receptor signaling pathways.

The protein KPNA2 plays a critical role in recognizing and importing proteins with specific nuclear localization signals, essential for regulating gene expression and controlling cellular functions, with its overexpression implicated in various cancers, influencing cell proliferation, migration, and invasion [31]. FOXM1 is a key regulator of the cell cycle involving in a variety of pathological processes [30,32]. Studying the interaction between KPNA2 and FOXM1 is of interest to us because KPNA2 is involved in transporting various proteins including FOXM1 into the cell nucleus and FOXM1 regulates the expression of KPNA2. This interaction can influence the regulation of gene expression and therefore cellular processes such as cell division and proliferation. We demonstrate that there is a strong correlation between the expression of FOXM1 and KPNA2. In addition, mRNA expression levels of FOXM1 downstream genes CCNB1 and CCNB2 are also significantly correlated with KPNA2 mRNA expression (Figure 1). Furthermore, the GSEA results showed that a high expression of KPNA2 was mainly associated with the G2/M phase transition of the cell cycle in the HALLMARK term. These data imply that KPNA2 is highly associated with the FOXM1 pathway in breast cancer. From TCGA and the GEO database, we proved that high expression of KPNA2, FOXM1, CCNB1, and CCNB2 were correlated with poor survival of BC patients, especially elevated levels of both KPNA2 and FOXM1 mRNA were significantly associated with poorer OS (Figure 4a, *p* = 0.0046) and DSS (Figure 4b, *p* = 0.006).

Interestingly, we discovered that the combination of high KPNA2 and low FOXM1 mRNA levels exhibited the most favorable outcomes in terms of overall survival (OS) (*p* = 0.0355) and disease-specific survival (DSS) (*p* = 0.0424) (Figure 5b) in HR-positive HER2-negative BC patients. KPNA2 plays a crucial role in estrogen receptor alpha (ERα) signaling and breast cancer progression. Studies have shown that KPNA2 is frequently overexpressed in breast cancer tissues compared to normal breast tissue and correlates with poor prognosis [10,33]. As a nuclear transport protein, KPNA2 is involved in the cytoplasmic retention of key DNA damage response proteins including BRCA1, RAD51, and CHK1 in breast cancer cells. High nuclear KPNA2 expression is associated with cytoplasmic localization of these proteins and their reduced nuclear expression [10]. Furthermore, KPNA2 has been identified as a novel target of miR-26a/b in the estrogen signaling pathway, where estrogen stimulation leads to the c-MYC-mediated suppression of miR-26a/b, resulting in increased KPNA2 expression. This estrogen/c-MYC/miR-26 axis promotes cell proliferation through KPNA2 and other targets like CHD1 and GREB1 [34]. The expression of KPNA2 is also regulated by E2F transcription factors, with E2F1 promoting and E2F7 inhibiting KPNA2 expression [31]. The interplay between FOXM1 and ERα plays a crucial role in breast cancer development and treatment response. Studies have shown that FOXM1 regulates ERα expression in breast cancer cells, and its target cyclin D1 correlates with ERα positivity [19,35]. FOXM1 has been identified as one of the estrogen-responsive genes, suggesting reciprocal regulation between FOXM1 and ERα [36]. While estrogen acts as a mitogenic factor primarily through ERα in mammary epithelial cells [37,38], ERα expression serves as a favorable prognostic marker, with approximately two-thirds of ERα-positive patients responding to anti-estrogen treatments like tamoxifen or aromatase inhibitors. These anti-estrogens typically induce cell cycle arrest at the G1/S phase and potentially lead to cell death [39]. However, about half of the initially responsive patients develop resistance and experience relapses after prolonged treatment [37,40]. The observation that ERα-positive breast cancer cells express elevated levels of FOXM1 protein suggests that FOXM1 expression is regulated by ERα, highlighting the importance of understanding FOXM1’s role in endocrine sensitivity and resistance [19].

Studies have demonstrated that KPNA2 expression shows a significant positive correlation with HER2 status in breast cancer, with notably higher expression levels observed in HER2-positive tumors compared to HER2-negative cases [41]. This elevated KPNA2 expression is associated with more aggressive tumor characteristics, including larger tumor size, higher grades, negative hormone receptor status, triple-negative phenotypes, and increased lymph node metastasis [11,41]. KPNA2 overexpression significantly impacts treatment outcomes, particularly contributing to resistance against targeted therapies such as trastuzumab, primarily through its role in the aberrant localization of DNA damage response proteins and cell cycle regulators [11]. High KPNA2 levels serve as an independent prognostic factor, correlating with shorter overall survival (OS) and event-free survival (EFS) in HER2-positive breast cancer patients, regardless of treatment intensity [41]. KNPA2 inhibitor was developing but the complexity of cancer biology and the need for more comprehensive studies on the efficacy and safety of KPNA2-targeted therapies contribute to this limited clinical use [42,43]. FOXM1 also demonstrates a significant positive correlation with HER2 expression, with studies showing that increased HER2 signaling enhances FOXM1 expression through the PI3K-Akt signaling pathway, establishing FOXM1 as a downstream target of HER2 [44,45]. Studies have shown that higher levels of HER2 are associated with increased expression of FOXM1 in both breast carcinoma cell lines and patient samples. Specifically, the overexpression of HER2 correlates with elevated FOXM1 mRNA levels, suggesting that HER2 may transcriptionally regulate FOXM1 expression [45]. This relationship manifests in more aggressive tumor characteristics, including larger tumor size, increased lymph vascular invasion, and higher rates of metastasis, with high FOXM1 expression correlating with poorer clinical outcomes and increased recurrence rates in HER2-positive breast cancers [46]. Notably, FOXM1 plays a critical role in treatment resistance, particularly against HER2-targeted therapies such as trastuzumab and lapatinib, where elevated FOXM1 levels enable cancer cells to evade therapeutic effects [18]. This intricate relationship positions FOXM1 as both a valuable prognostic marker and a potential therapeutic target.

The multifaceted role of FOXM1 in BC biology, elucidating its regulation of ER-alpha production and interaction with ER-beta1, potentially impacting treatment response [43,44,45]. Increased FOXM1 levels in Luminal subtype BC patients receiving chemotherapy or tamoxifen are associated with poorer outcomes, suggesting a link to treatment resistance or noncompliance [27,46,47]. Another study found that high levels of 14-3-3ζ in breast tumors, particularly in Luminal B subtype, correlate with endocrine resistance [47]. 14-3-3ζ regulates FOXM1, a key player in cell division, and promotes cell survival and resistance to endocrine therapies. Targeting 14-3-3ζ and its associated proteins, like FOXM1, could potentially reverse endocrine resistance and reduce BC recurrence risk. FOXM1 plays a pivotal role in cancer progression, including BC, by promoting unhindered proliferation, tumorigenesis, cell migration, epithelial–mesenchymal transition (EMT), and metastasis [48]. Drug therapies targeting FOXM1 have shown promise in slowing cancer progression and inducing apoptosis. Inhibitors like thiostrepton [49], siomycin A [50], and the ARF26-44 peptide [51] have demonstrated efficacy in suppressing FOXM1’s downstream gene targets and inhibiting tumor growth. This indicates the potential therapeutic value of FOXM1 for BC treatment in the future.

Our study has several limitations. First, the rationale for analyzing the relationship between KPNA2 and FOXM1 based on the existing literature is relatively weak. Second, the correlation analysis presented in Figure 1 and Figure 2 focuses on the associations between genes and between individual genes and breast cancer subtypes, yielding consistent results across three different cohorts. However, the follow-up duration in our series is short, limiting our ability to demonstrate that HR+/HER2- breast cancer patients with high KPNA2 but low FOXM1 mRNA levels exhibit significantly better survival outcomes beyond the TCGA cohorts. Lastly, the sample size of the GSE7390 validation cohort is too small to establish a robust multi-cohort validation of the findings.

## 5. Conclusions

In conclusion, the findings suggest that KPNA2 and FOXM1 play a crucial role in breast cancer progression by positively correlating with key genes involved in cell cycle regulation and molecular subtypes. High expression of KPNA2, along with FOXM1, CCNB1, and CCNB2, is associated with poorer survival outcomes in BC patients. Notably, a combination of high KPNA2 and low FOXM1 levels indicates a particularly favorable prognosis in HR-positive HER2-negative BC patients. This is the first paper that demonstrates high KPNA2/low FOXM1 predicts better survival in HR+/HER2- breast cancer. These results highlight the potential significance of combining KPNA2 and FOXM1 as prognostic markers and therapeutic targets in BC management. Due to the lack of targeted KPNA2 drugs till now, targeted FOXM1 drugs may play some roles in BC treatment in the future, especially for HR-positive HER2-negative BC patients.

## Figures and Tables

**Figure 1 cancers-17-00671-f001:**
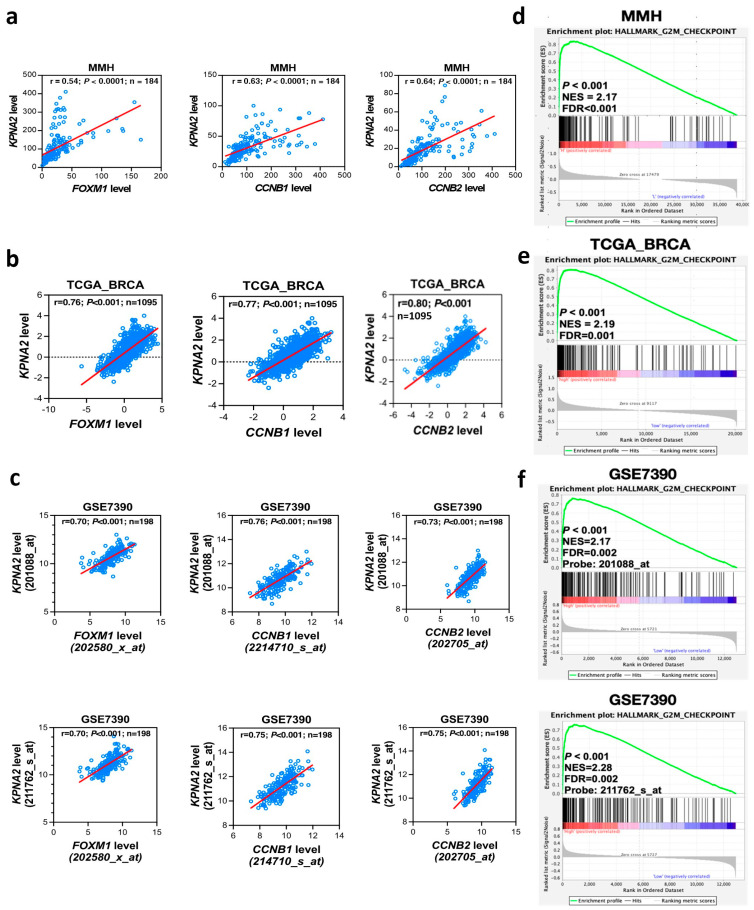
Functional association and correlation of the expression of KPNA2 with FOXM1 and its downstream effectors, CCNB1 and CCNB2. (**a**–**c**) Pearson correlation assessment of transcriptional co-regulation between KPNA2 and FOXM1(left), CCNB1(middle), and CCNB2 (right). Corresponding correlation coefficients, r; *p* values and cohort size (n) are presented. Significant and positive linear correlations were observed for each pair-wise analysis in the MMH dataset (**a**), which were also confirmed in the TCGA_BRCA (**b**) and GSE7390 (**c**) datasets for external validation across platforms. Two Affymetrix microarray probes (201088_at and 211762_s_at), both interrogating KPNA2, yielded consistent results in GSEA and correlation analysis. (**d**–**f**) Gene Set Enrichment Analysis (GSEA) of the MMH (**d**), TCGA_BRCA (**e**) or GSE7390 (**f**) transcriptome datasets revealed that the priori-defined “Hallmark_G2M_CHECKPOINT” 200-genes set (black bars) in the Molecular Signatures Database (MSigDB) was overrepresented in the gene expression positively (red region) correlated with the KPNA2 transcript levels in the respective cohorts. The magnitude of overrepresentation is reflected by the maximal enrichment score (ES) based on a weighted running sum (green curve) and its significance by the empirical *p*-value based on 1000 permutations of the phenotype class labels. Patient samples were sorted based on the *KPNA2* transcript levels to give two phenotype classes of “KPNA high” and “KPNA low”, comprising samples with *KPNA2* expression exceeding the cohort median, and those with expression levels below the median, respectively. The default weighted Signal2Noise setting was applied to assign the ranking metric scores (gray bar plot) based on the degree of correlation with phenotype rankings. The ES was subsequently normalized (NES) to account for the size of gene set among all others in MSigDB, and a false discovery rate (FDR) determined based on the NES to control for false positive findings arising from multiple hypothesis testing.

**Figure 2 cancers-17-00671-f002:**
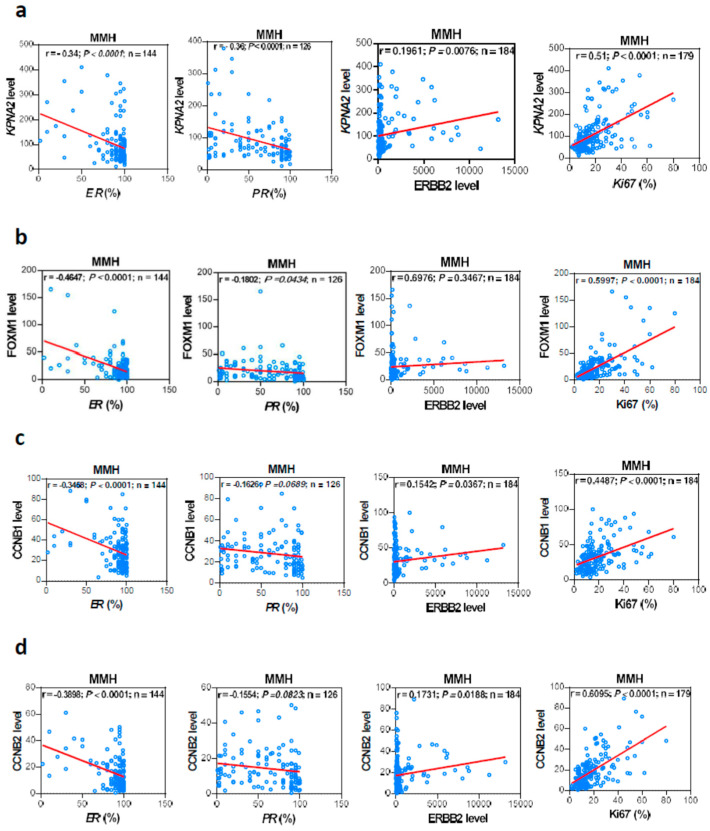
Correlation of prognostic genes expression with receptors status and the Ki67 proliferation index. KPNA2 (**a**), FOXM1(**b**), CCNB1 (**c**), and CCNB2 (**d**) transcript levels in MMH transcriptome analysis were plotted against the clinicopathological assessment of ER-positive, PR-positive, Ki67-positive fractions, or the HER2 (ERBB2) transcript levels. Corresponding Pearson’s correlation coefficient and *p* values are presented.

**Figure 3 cancers-17-00671-f003:**
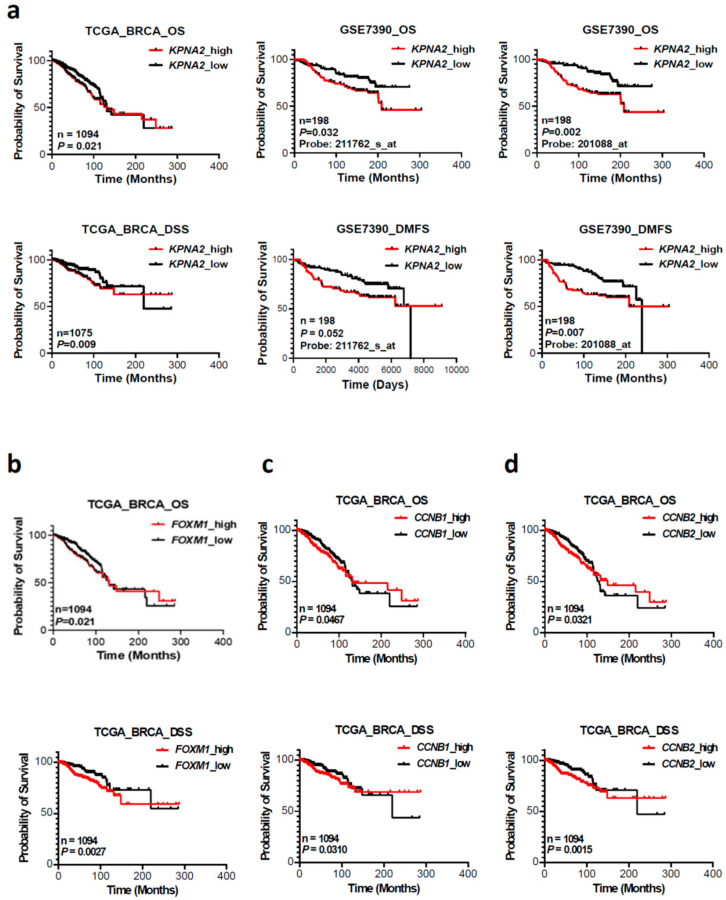
External validations of the four prognostic genes by Kaplan–Meier survival analysis of subgroups based on transcript levels. (**a**). TCGA or GSE7390 dataset patients were split into KPNA2_low and KPNA2_high subgroups according to the cohort median of KPNA2 transcript levels. The overall survival (OS) and disease-specific survival (DSS) and metastasis-free survival (MFS) were analyzed to assess the prognostic capabilities of KPNA2 expression. (**b**–**d**). The individual associations of FOXM1, CCNB1, and CCNB2 with OS and DSS in the TCGA dataset were similarly assessed by the K-M plots.

**Figure 4 cancers-17-00671-f004:**
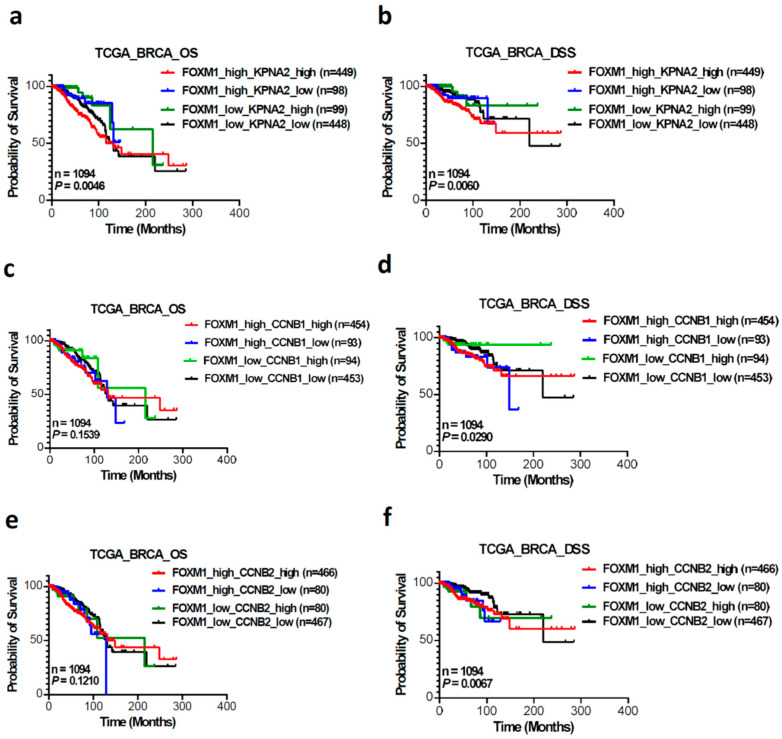
The prognostic value of combining FOXM1 and KPNA2 and the CCNB1 and CCNB2 mRNA levels of BC patients in the TCGA dataset were assessed by Kaplan–Meier survival analysis. Among the 1094 patients, four groups were defined based on KPNA2 and FOXM1 expression: high FOXM1 and high KPNA2, high FOXM1 and low KPNA2, low FOXM1 and high KPNA2, and low FOXM1 and low KPNA2. High levels of both KPNA2 and FOXM1 mRNA were significantly associated with poorer overall survival (OS) (**a**) and disease-specific survival (DSS) (**b**). Furthermore, combined expression of FOXM1 with CCNB1 or CCNB2 also showed poorer DSS. Specifically, the FOXM1 and CCNB1 group exhibited poorer DSS (**d**) but no significant effect on OS (**c**). Similarly, the FOXM1 and CCNB2 group showed a poorer DSS (**f**) with no significant impact on OS (**e**).

**Figure 5 cancers-17-00671-f005:**
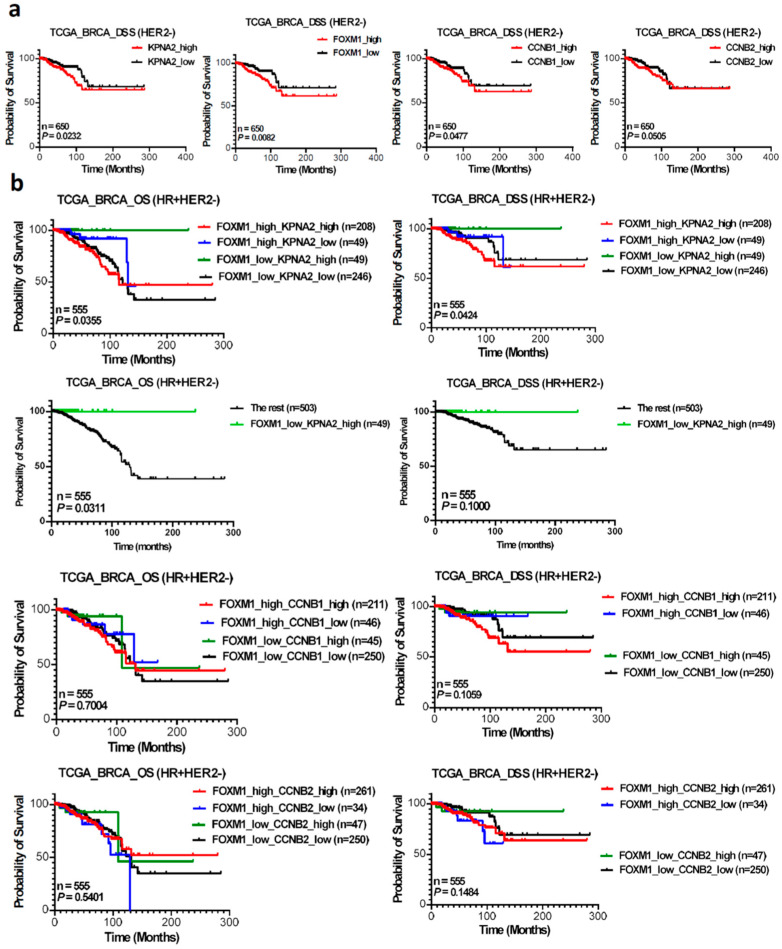
KPNA2, FOXM1, CCNB1, and CCNB2 expression levels and their association with OS and DSS in the TCGA dataset, stratified by ER, PR, and HER2 status using Kaplan–Meier plots. (**a**) The outcome prognostication is concordantly significant for disease-specific survival (DSS) and only in the HER2-negative subgroup of invasive breast cancer in the TCGA dataset. (**b**) The prognostic potential of combining KPNA2 and FOXM1 mRNA levels was evaluated in the HR-positive HER2-negative subset of the TCGA dataset. KPNA2 in combination with FOXM1 maintained the significant stratification of patients in terms of both overall survival (OS) (Panel (**b**), uppermost and second left) and DSS (Panel **b**, upper rightmost) as determined by the K-M survival analysis. In contrast, combing FOXM1 and CCNB1 (Panel (**b**), third) and combing FOXM1 and CCNB2 (panel (**b**), bottom) failed to retain a significant separation of OS and DSS.

**Table 1 cancers-17-00671-t001:** Clinical data distribution and association with KPNA2 transcript levels in the cohort of women with breast cancer whose primary tumors were examined by RNAseq for transcriptome profiling.

	Entire Cohort (n = 184)	Patients with Low * KPNA2 (n = 92)	Patients with High * KPNA2 (n = 92)	*p*-Value
**Age at diagnosis**, median (min-max)	53 (27–93)	51 (27–93)	56 (37–74)	*p* = 0.8357
**Receptor status**				
ER-positive	144 (78%)	80 (87%)	64 (70%)	*p* = 0.0069
PR-positive	126 (68%)	76(83%)	50 (54%)	*p* < 0.0001
Her2-positive	40 (22%)	11 (12%)	29 (32%)	*p* = 0.0022
**Ki67 > 14**	84 (46%)	22 (24%)	62 (67%)	*p* < 0.0001
**Intrinsic subtype**				*p* = 0.0003
Luminal A	82 (45%)	60 (65%)	22 (24%)	
Luminal B	41 (22%)	13 (14%)	28 (30%)	
Luminal Her2	22 (12%)	6 (7%)	16 (17%)	
Her2	16 (9%)	4 (4%)	12 (13%)	
Triple negative	20 (11%)	6 (7%)	14 (15%)	
**Pathology type**				*p* = 0.5311
Ductal	149 (81%)	71 (77%)	78 (85%)	
Lobular	8 (4%)	6 (7%)	2 (2%)	
Papillary	9 (5%)	6 (7%)	3 (3%)	
Mucinous	6 (3%)	3 (3%)	3 (3%)	
Others	12 (7%)	6 (7%)	6 (7%)	
**Tumor grade, overall**				*p* < 0.0001
Grade I	27 (15%)	23 (25%)	4 (4%)	
Grade II	119 (65%)	62 (67%)	57 (62%)	
Grade III	26 (14%)	3 (3%)	23 (25%)	
Not graded	12 (7%)	4 (4%)	8 (9%)	
**TNM stage, pT**				*p* = 0.0101
T1	82 (45%)	52 (57%)	30 (33%)	
T2	80 (43%)	33 (36%)	47 (51%)	
T3 and above	18 (10%)	7 (8%)	11 (12%)	
**TNM stage, pN**				*p* = 0.8348
N0	119 (65%)	61 (66%)	58 (63%)	
N1	52 (28%)	24 (26%)	28 (30%)	
N2 and above	13 (7%)	7 (8%)	6 (7%)	
**TNM stage, pM**				*p* = 0.2122
M0 (up to last follow-up date)	173 (65%)	89 (65%)	84 (65%)	
M1	11 (65%)	3 (65%)	8 (65%)	
**Stage by AJCC ** ver. 8**				*p* = 0.0107
Stage I	110 (60%)	65 (71%)	45 (49%)	
Stage II	50 (27%)	19 (21%)	31 (34%)	
Stage III/IV	24 (13%)	8 (9%)	16 (17%)	
**Surgery type**				*p* = 0.5267
Core needle biopsy	11 (6%)	3 (3%)	8 (9%)	
Modified radical mastectomy	6 (3%)	4 (4%)	2 (2%)	
Partial mastectomy	125 (68%)	62 (67%)	63 (68%)	
Simple Mastectomy	31 (17%)	17 (18%)	14 (15%)	
Total Mastectomy	9 (5%)	5 (5%)	4 (4%)	

* Low or high KPNA2: KPNA2 transcript levels lower or higher than the cohort median. ** AJCC: American Joint Commission of Cancer.

**Table 2 cancers-17-00671-t002:** Patient characteristics of the MMH cohort. Clinicopathological data associated with the KPNA2/FOXM1/CCNB1/CCNB2 transcript levels of the MMH cohort of women with breast cancer whose primary tumors were examined by RNAseq for transcriptome profiling.

	Entire Cohort	*p*-Values			
	(n = 184)	Low * Expression vs. High * Expression			
		KPNA2	FOXM1	CCNB1	CCNB2
**Receptor status**					
ER-positive	144 (78%)	*p* = 0.0069	*p* = 0.0069	*p* = 0.0194	*p* = 0.0021
PR-positive	126 (68%)	*p* < 0.0001	*p* < 0.0001	*p* = 0.0067	*p* < 0.0001
Her2-positive	40 (22%)	*p* = 0.0022	*p* = 0.0022	*p* = 0.0079	*p* = 0.0022
**Ki67 > 14**	85 (46%)	*p* < 0.0001	*p* < 0.0001	*p* < 0.0001	*p* < 0.0001
**Intrinsic subtype**		*p* = 0.0003	*p* < 0.0001	*p* < 0.0001	*p* < 0.0001
Luminal A	82 (45%)				
Luminal B Her2−	41 (22%)				
Luminal B Her2+	22 (12%)				
Her2	16 (9%)				
Triple negative	20 (11%)				
**Pathology type**		*p* = 0.4765	*p* = 0.0120	*p* = 0.2618	*p* = 0.1016
Ductal	120 (65%)				
Lobular	8 (4%)				
Papillary	7 (4%)				
Mucinous	6 (3%)				
Others	43 (23%)				
**Tumor grade, overall**		*p* < 0.0001	*p* < 0.0001	*p* < 0.0001	*p* < 0.0001
Grade I	27 (15%)				
Grade II	119 (65%)				
Grade III	26 (14%)				
Not graded	4 (4%)				
**TNM stage, pT**		*p* = 0.0390	*p* = 0.4377	*p* = 0.4579	*p* = 0.0997
T1	82 (45%)				
T2	76 (41%)				
T3 and above	6 (3%)				
**TNM stage, pN**		*p* = 0.7611	*p* = 0.2854	*p* = 0.4806	*p* = 0.7046
N0	120 (65%)				
N1	40 (22%)				
N2 and above	9 (5%)				
**TNM stage, pM**		*p* = 0.2122	*p* = 0.2122	*p* = 0.2122	*p* = 0.0576
M0 (up to last follow-up date)	173 (94%)				
M1	11 (6%)				
**Stage by AJCC ** ver. 8**		*p* = 0.0107	*p* = 0.0265	*p* = 0.0143	*p* = 0.0440
Stage I	110(60%)				
Stage II	50 (27%)				
Stage III/IV	24 (13%)				
**Surgery type**		*p* = 0.5267	*p* = 0.2946	*p* = 0.1923	*p* = 0.0971
Core needle biopsy	11 (6%)				
Modified radical mastectomy	6 (3%)				
Partial mastectomy	121 (66%)				
Simple mastectomy	31 (17%)				
Total mastectomy	9 (5%)				

* Low or high KPNA2/FOXM1/CCNB1/CCNB2: KPNA2/FOXM1/CCNB1/CCNB2 transcript levels lower or higher than the cohort median for the respective transcript. ** AJCC: American Joint Commission of Cancer.

## Data Availability

The data presented in this study are available on request from the corresponding authors. The data are not publicly available due to the ethics approval agreement.

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
