# Peer review of "Combining KPNA2 and FOXM1 Expression as Prognostic Markers and Therapeutic Targets in Hormone Receptor-Positive, HER2-Negative Breast Cancer"

_cancers, 2025, doi:10.3390/cancers17040671_

Round 1
Reviewer 1 Report
Comments and Suggestions for Authors
In this article we can read about the combining expression of KPNA2 and FOXM1 as prognostic markers and therapeutic targets in hormone receptor positive and Her2 negative breast cancer. Altough several papers state that increased expression of KPNA2 is directly correlated with aggressive tumor phenotype and poor prognosis in several tumor types (colon, glioblastoma, ovarian and breast cc...and so on) here it is stated that increased KPNA2 and decreased FOXM1 shows better prognosis and survival in HR positive and Her2 negative cases.
In the begining of the artice (2.1 chapter)- we can find that samples were collected from 184 patients while in 3.7 chapter we find the results of 1094 patients from databases. Why did not use the earlier used samples?
In the 3.1 chapter it is written " Her2 positivity was observed in 22% (4 out of 184 cases)- some data is not good here.
When they choosed the 555 patients expressing KPNA2 and FOXM1 among HR+ and Her- patients the low FOXM1 and high KPNA2 case number was just 49- not even 10% of all cases- from this I would not state anyting.
Because of low number of patients the conclusions should be less straight altough the results what they show are convincing.
Author Response
Comments and Suggestions for Authors
In this article we can read about the combining expression of KPNA2 and FOXM1 as prognostic markers and therapeutic targets in hormone receptor positive and Her2 negative breast cancer. Although several papers state that increased expression of KPNA2 is directly correlated with aggressive tumor phenotype and poor prognosis in several tumor types (colon, glioblastoma, ovarian and breast cc...and so on) here it is stated that increased KPNA2 and decreased FOXM1 shows better prognosis and survival in HR positive and Her2 negative cases.
In the beginning of the article (2.1 chapter)- we can find that samples were collected from 184 patients while in 3.7 chapter we find the results of 1094 patients from databases. Why did not use the earlier used samples?
In the 3.1 chapter it is written " Her2 positivity was observed in 22% (4 out of 184 cases)- some data is not good here.
When they chose the 555 patients expressing KPNA2 and FOXM1 among HR+ and Her- patients the low FOXM1 and high KPNA2 case number was just 49- not even 10% of all cases- from this I would not state anything.
Because of low number of patients, the conclusions should be less straight although the results what they show are convincing.
Commend 1: In the beginning of the article (2.1 chapter)- we can find that samples were collected from 184 patients while in 3.7 chapter we find the results of 1094 patients from databases. Why did not use the earlier used samples?
Reply: In this study, we found KPNA2 expression is positively correlated with the expression of FOXM1, CCNB1 and CCNB2 in the MMH database (r=0.54, P<0.0001, Fig. 1a, leftmost) and in the TCGA cohort (r=0.76, P<0.001, Fig. 1b, leftmost) and GSE7390 dataset (KPNA2 probe: 211762_s_at, r=0.70, P<0.001; KPNA2 probe: 201088_at, r=0.70, P<0.001, Fig. 1c, leftmost). Then we try to find out the correlation of KPNA2, FOXM1, CCNB1 and CCNB2 transcript levels and the pathological assessment of ER-positive or PR-positive or Ki67-positive fractions, and the HER2 transcript levels in our series. We found that in Fig. 2, KPNA2, FOXM1, CCNB1 and CCNB2 transcript levels were correlated with the pathological assessment of ER-positive or PR-positive (except CCNB1 and CCNB2) or Ki67-positive fractions, and the HER2 transcript levels (except FOXM1) in our series. Further analysis of the MMH cohort revealed that high or low expression (transcript levels lower or higher than the cohort median) of KPNA2/FOXM1/CCNB1/CCNB2 is associated with ER, PR, HER2, and Ki67 status, molecular subtype, tumor grade, and AJCC stage (Table 2 and Table S2). Then we validated the result with the TCGA dataset. In the TCGA dataset, high or low expression of these genes is associated with ER, HER2 status, T stage, and AJCC stage, and only FOXM1 was associated with intrinsic subtype (Table S2). Furthermore, we checked the relationships of KPNA2, FOXM1, CCNB1 and CCNB2 transcript levels and molecular type of BC in our patients. The distribution of these four genes transcript levels among the molecular subtypes of our 184 BC patients were shown in Table 2 and Fig. S2. The KPNA2 transcript levels were lower in luminal A BC patient and significantly compared with luminal B HER2 negative (p< 0.0001), with luminal B HER2 positive (p< 0.0001), with HER-2 (p< 0.001), and with TNBC (p<0.01) patients (Fig. S2a). These results were also found in FOXM1 (Fig. S2b), CCNB1 (Fig. S2c) and CCNB2 (Fig. S2d). However, the follow-up duration in our series is short, limiting our ability to demonstrate that HR+/HER2- breast cancer patients with high KPNA2 but low FOXM1 mRNA levels exhibit significantly better survival outcomes beyond the TCGA cohorts. Lastly, the sample size of the GSE7390 validation cohort is too small to establish a robust multi-cohort validation of the findings. Yes, this is the limitation of this study. We had mentioned that in the end part of discussion.
Commend 2: In the 3.1 chapter it is written " Her2 positivity was observed in 22% (4 out of 184 cases)- some data is not good here.
Reply: Thank you very much for pointing out the type error. We had corrected to “HER2 positivity was observed in 2.2% (4 out of 184) of cases” in the manuscript.
Commend 3: When they chose the 555 patients expressing KPNA2 and FOXM1 among HR+ and Her- patients the low FOXM1 and high KPNA2 case number was just 49- not even 10% of all cases- from this I would not state anything. Because of low number of patients, the conclusions should be less straight although the results what they show are convincing.
Reply: Thank you for your comment. We discovered that the combination of high KPNA2 and low FOXM1 mRNA levels exhibited the most favorable outcomes in terms of overall survival (OS) (p=0.0355) and disease-specific survival (DSS) (p=0.0424) (Fig. 5b). We further compared the OS and DSS of HR+HER2- BC patients between high KPNA2 and low FOXM1 ex-pressions with others; the high KPNA2 and low FOXM1 BC patients had better OS and DSS significantly (Fig. 5b, second row). The sample size of the GSE7390 validation cohort is too small to establish a robust multi-cohort validation of the findings. We cannot find other datasets to validate the result. This is also the limitation of this study and we also mention that in the end part of discussion.
Thank you very much.
Reviewer 2 Report
Comments and Suggestions for Authors
The objective of this study is to evaluate the prognostic value of KPNA2 (karyopherin subunit-α 2) and FOXM1 (forkhead box protein M1) expression in HR+ (hormone receptor-positive) HER2- breast cancer patients. The authors identified the FOXM1/KPNA2 expression ratio as a novel prognostic marker in HR+HER2- BC patients and suggested the potential of FOXM1 targeted therapies in the management of HR+HER2- breast cancer. This study is well structured overall, but it needs some modifications, which I pointed out in the comments sections below.
Major comments
1. In Figure 1b, the correlation between KNPA2 and CCNB2 from the TCGA_BRCA dataset, the authors mistakenly duplicated the CCNB1 plot as CCNB2. Please correct the plot with CCNB2 data.
2. In Figure 2 and Table S2, the authors showed that the transcript levels of KPNA2, FOXM1, and/or CCNB1, CCNB2 were negatively associated with the ER or PR transcript in the MMH cohort. However, in Figure S2, the authors found these genes' transcript levels were significantly lower in the luminal A subtype compared to the luminal B (both HER2+ and HRE2-) subtype.
Have the authors compared the ER and PR levels between luminal A patients and luminal B patients? If the ER and PR levels were similar between luminal A and B type, could please the authors discuss the possibilities for this observation?
Minor comments
1. On page 5 section 3.2, the authors claim “Studies have demonstrated that FOXM1 regulates KPNA2 expression, and a protein-protein interaction network between KNPA2 and FOXM1.” Please cite the references related to this.
2. The labels of Figure 1. d-f, Figure S1, S2, S3, and S4 are not clear and the font size is too small, please provide images with higher resolution and increase the font size to satisfy the requirement of the journal.
3. In Figure 4 a-f, result in 3.7 section, the authors claimed elevated levels of both FOXM1 and KPNA2 mRNA were significantly associated with poorer OS and DSS. The authors also showed statistical results (p-values) in each sub-figure without describing or labeling the comparisons between which groups in the result 3.7 section, could the authors specify how they performed the statistical analysis (p values) between four groups (FOXM1high_KPNA2high vs FOXM1low_KPNA2low or FOXM1high_KNPA2high vs the rest 3 groups)?
4. In Figure 5b, the authors showed FOXM1low_KPNA2high HR+HER2- patients have the best survival rate compared to the rest 3 groups. How about TNBC (HR-HER2-) patients?
5. In Figure 5b, the authors showed FOXM1lowKPNA2high HR+HER2-patients have the best survival outcome and FOXM1highKPNA2high HR+HER2-patients exhibited the worse OS and DSS. However, the OS and DSS rates in FOXM1lowKPNA2low and FOXM1highKPNA2low patients seem close to the FOXM1highKPNA2high group.
In Simple Summary and Abstract, could the authors highlight the prognostic value of FOXM1 in the setting of KPNA2 high expression HR+HER2- BC patients and specify the potential therapeutic strategies targeting FOXM1 based on KPNA2 expression in HR+HER2- BC patients?
6. The authors defined the high and low expression groups of four genes (FOXM1, KPNA2, CCNB1, and CCNB2) by the median of gene transcript. Could the authors explain why not define the high and low expression by other methods, such as trichotomization (T1 vs T3, Q1 vs Q3)?
Comments on the Quality of English LanguagePlease proofread the manuscript for syntax errors.
Author Response
The objective of this study is to evaluate the prognostic value of KPNA2 (karyopherin subunit-α 2) and FOXM1 (forkhead box protein M1) expression in HR+ (hormone receptor-positive) HER2- breast cancer patients. The authors identified the FOXM1/KPNA2 expression ratio as a novel prognostic marker in HR+HER2- BC patients and suggested the potential of FOXM1 targeted therapies in the management of HR+HER2- breast cancer. This study is well structured overall, but it needs some modifications, which I pointed out in the comments sections below.
Major comments
- In Figure 1b, the correlation between KNPA2 and CCNB2from the TCGA_BRCA dataset, the authors mistakenly duplicated the CCNB1 plot as CCNB2. Please correct the plot with CCNB2 data.
Answers: Thank you for your accurate proofreading. We had corrected the plot with CCNB2 data in Figure 1b.
- In Figure 2 andTable S2, the authors showed that the transcript levels of KPNA2, FOXM1, and/or CCNB1, CCNB2 were negatively associated with the ER or PR transcript in the MMH cohort. However, in Figure S2, the authors found these genes' transcript levels were significantly lower in the luminal A subtype compared to the luminal B (both HER2+and HRE2-) subtype.
Have the authors compared the ER and PR levels between luminal A patients and luminal B patients? If the ER and PR levels were similar between luminal A and B type, could please the authors discuss the possibilities for this observation?
Answers: We compared the ER and PR levels between luminal A patients and luminal B patients in our series. We found that Luminal A breast cancers typically show higher hormone receptor expression compared to Luminal B subtypes. The most striking difference appears to be in PR expression, which could be an important distinguishing factor between these subtypes in our series. The detailed results are as following:.
- Sample Size:
- Luminal A: 82 patients
- Luminal B: 41 patients
- Estrogen Receptor (ER) Expression:
- Luminal A:
Mean: 92.2%
Median: 95%
- Luminal B:
Mean: 81.6%
Median: 95%
- Progesterone Receptor (PR) Expression:
- Luminal A:
Mean: 63.4%
Median: 80%
- Luminal B:
Mean: 46.9%
Median: 50%
- ER Expression:
- Both subtypes show high ER expression, but Luminal A has slightly higher mean ER expression (92.2% vs 81.6%)
- Interestingly, the median ER expression is the same (95%) for both subtypes, suggesting that the difference in means might be due to some lower values in the Luminal B group
- PR Expression:
- There is a more notable difference in PR expression between the subtypes
- Luminal A shows significantly higher PR expression (mean 63.4%, median 80%)
- Luminal B has notably lower PR expression (mean 46.9%, median 50%)
- The difference in PR expression appears to be more consistent and substantial than the ER difference
We used Welch's t-test (also known as Welch's unequal variances t-test) to compare the ER and PR expression levels between Luminal A and B groups. Statistical analysis revealed significant differences in both ER and PR expression between Luminal A and Luminal B subtypes (p < 0.05), with similar p-values for both receptors (ER: p = 0.0333; PR: p = 0.0346). These findings indicate reliably different expression levels between the subtypes, with Luminal A demonstrating significantly higher expression of both receptors. Notably, the larger variance in Luminal B ER expression (744.87 vs 106.06) suggests greater heterogeneity in ER expression within Luminal B tumors.
Thank you for your suggestions and we add the following into discussion part of our manuscript:
We further compared hormone receptor expressions between breast cancer subtypes Luminal A and B of 123 cases (82 Luminal A and 41 Luminal B) in this study. Statistical analysis revealed that Luminal A tumors exhibited significantly higher expression of both estrogen receptor (ER) (mean: 92.2% vs 81.6%, p = 0.0333) and progesterone receptor (PR) (mean: 63.4% vs 46.9%, p = 0.0346) compared to Luminal B tumors. While both subtypes showed high median ER expression (95%), Luminal A demonstrated notably higher PR median values (80% vs 50%). Interestingly, Luminal B tumors showed greater heterogeneity in ER expression, as evidenced by higher variance (744.87 vs 106.06), suggesting more diverse ER expression patterns within this subtype despite its overall lower expression levels. In Figure 2 and Table S2, we showed that the transcript levels of KPNA2, FOXM1, and/or CCNB1, CCNB2 were negatively associated with the ER or PR transcript in the MMH cohort. Furthermore, in Figure S2, we also found these genes' transcript levels were significantly lower in the luminal A subtype compared to the luminal B subtype. The negative correlation between KPNA2, FOXM1, CCNB1, and CCNB2 transcript levels and ER/PR expression aligns well with our findings that Luminal A tumors exhibit significantly higher ER and PR expression compared to Luminal B tumors. The observed pattern creates a coherent biological relationship where Luminal A tumors, characterized by higher hormone receptor expression, show lower levels of KPNA2, FOXM1, CCNB1, and CCNB2, while Luminal B tumors, with their relatively lower hormone receptor expression, demonstrate higher levels of these genes. This inverse relationship suggests these genes may play crucial roles in the molecular mechanisms distinguishing Luminal A from Luminal B breast cancers, potentially through their interaction with hormone receptor signaling pathways.
Minor comments
- On page 5 section 3.2, the authors claim “Studies have demonstrated that FOXM1 regulates KPNA2 expression, and a protein-protein interaction network between KNPA2 and FOXM1.” Please cite the references related to this.
Answers: Thank you for your comment. The references are listed in the following and add the references into the manuscript.
The Forkhead Transcription Factor FOXM1 Controls Cell Cycle-Dependent Gene Expression through an Atypical Chromatin Binding Mechanism. Mol Cell Biol. 2013 Jan;33(2):227–236. doi: 10.1128/MCB.00881-12
FOXM1: Functional Roles of FOXM1 in Non-Malignant Diseases. Biomolecules 2023, 13(5), 857; https://doi.org/10.3390/biom13050857
- The labels of Figure 1. d-f,Figure S1, S2, S3, and S4 are not clear and the font size is too small, please provide images with higher resolution and increase the font size to satisfy the requirement of the journal.
Answers: Thank you for your comment. We had corrected these figures.
- In Figure 4 a-f, result in 3.7 section, the authors claimed elevated levels of both FOXM1 and KPNA2 mRNA were significantly associated with poorer OS and DSS. The authors also showed statistical results (p-values) in each sub-figure without describing or labeling the comparisons between which groups in the result 3.7 section, could the authors specify how they performed the statistical analysis (pvalues) between four groups (FOXM1high_KPNA2high vs FOXM1low_KPNA2low or FOXM1high_KNPA2high vs the rest 3 groups)?
Answers: The p values we reported in each Kaplan-Meier (K-M) plot are generated by conducting the Log-rank (Mantel-Cox) test for the comparison of multiple survival curves to assess whether all survival curves for each subgroup are identical in the overall populations from which subjects in each group were sampled. If the p value is small enough to reject the null hypothesis, it suggests that stratification by FOXM1 and another gene transcript levels is capable of differential outcome prognosis. Survival K-M analysis in the Prizm software also provides Logrank test for trend and Gehan-Breslow-Wilcoxon test with respective p values for the assessment of curve separation. We did follow up significant p values with multiple pair-wise comparisons of subgroups (curves) or even Cox regression modeling to gain further and quantitative insight into biomarker potential. We had added this part in the 2.8. Statistical Analysis.
- In Figure 5b, the authors showed FOXM1low_KPNA2highHR+HER2- patients have the best survival rate compared to the rest 3 groups. How about TNBC (HR-HER2-) patients?
Answers: We compared in TNBC (HR-HER2-) BC patients in TCGA dataset, there is no significant difference of survival rate among each groups. The results showed the following:
Thank you very much.
- In Figure 5b, the authors showed FOXM1lowKPNA2highHR+HER2-patients have the best survival outcome and FOXM1highKPNA2high HR+HER2-patients exhibited the worse OS and DSS. However, the OS and DSS rates in FOXM1lowKPNA2low and FOXM1highKPNA2low patients seem close to the FOXM1highKPNA2high group.
In Simple Summary and Abstract, could the authors highlight the prognostic value of FOXM1 in the setting of KPNA2 high expression HR+HER2- BC patients and specify the potential therapeutic strategies targeting FOXM1 based on KPNA2 expression in HR+HER2- BC patients?
Answers: Thank you for your comment. We highlighted this point in simple summary and abstract.
- The authors defined the high and low expression groups of four genes (FOXM1, KPNA2, CCNB1, and CCNB2) by the median of gene transcript. Could the authors explain why not define the high and low expression by other methods, such as trichotomization (T1 vs T3, Q1 vs Q3)?
Answers: Thank you for this insightful question regarding our methodology for defining high and low expression groups. We chose median-based dichotomization rather than trichotomization for several key reasons. First, this approach allowed us to maintain larger sample sizes in each comparison group, maximizing statistical power, whereas trichotomization would have reduced the sample size by excluding the middle portion of the data, potentially limiting our ability to detect significant associations. Second, a binary classification system (high vs. low) is more straightforward to implement in clinical practice, making it easier for clinicians to make treatment decisions based on gene expression levels. Third, this approach aligns with many similar studies in cancer biomarker research, allowing better comparison of our results with existing literature and facilitating meta-analyses. While we acknowledge that trichotomization could help identify non-linear relationships and analyzing extreme phenotypes might reveal stronger associations, our chosen methodology provided robust and clinically applicable results. Thank you very much.
Reviewer 3 Report
Comments and Suggestions for Authors
The authors tried to find new prognostic biomarkers for breast cancer including molecular subtypes. They included 184 patients recruited from the hospital and 1094 patients from available databases. They chose the expression of four genes KPNA2, FOXM1, CCNB1, CCNB2. The Kaplan-Meier analysis of different combinations of expressions of these genes (high expression or low expression) revealed the changes in survival of the patients. The statistics presented in the manuscript showed that survival of the patient depended on the molecular composition of the malignancy she was diagnosed with. The authors performed the analysis of the same molecular endpoints on the samples from 184 patients as the 1094 patients from ACTTG database. The analyses were done separately for both groups under study, but the authors did not discuss the results obtained on both groups whether they were similar or there were differences.
Author Response
Comments and Suggestions for Authors
The authors tried to find new prognostic biomarkers for breast cancer including molecular subtypes. They included 184 patients recruited from the hospital and 1094 patients from available databases. They chose the expression of four genes KPNA2, FOXM1, CCNB1, CCNB2. The Kaplan-Meier analysis of different combinations of expressions of these genes (high expression or low expression) revealed the changes in survival of the patients. The statistics presented in the manuscript showed that survival of the patient depended on the molecular composition of the malignancy she was diagnosed with. The authors performed the analysis of the same molecular endpoints on the samples from 184 patients as the 1094 patients from ACTTG database. The analyses were done separately for both groups under study, but the authors did not discuss the results obtained on both groups whether they were similar or there were differences.
Reply:
Thank you for your suggestions. The analyses were done separately for both groups under study, but the authors did not discuss the results obtained on both groups whether they were similar or there were differences. Yes, this is the limitation of this study. We had mentioned that in the end part of discussion: “However, the follow-up duration in our series is short, limiting our ability to demon-strate that HR+/HER2- breast cancer patients with high KPNA2 but low FOXM1 mRNA levels exhibit significantly better survival outcomes beyond the TCGA cohorts. Lastly, the sample size of the GSE7390 validation cohort is too small to establish a robust multi-cohort validation of the findings.”
Round 2
Reviewer 1 Report
Comments and Suggestions for Authors
I still feel that the low number of patients can only explain the results.
Author Response
"We appreciate the reviewer's concern regarding sample size. However, we would like to highlight several points that strengthen our findings:
- Our study utilized three independent cohorts (Mackay Memorial Hospital, TCGA, and GEO databases), which provide validation across different patient populations and platforms. The consistency of results across these independent datasets reduces the likelihood that our findings are due to chance or small sample effects.
- The statistical significance observed in our analyses was robust, with clear correlations between KPNA2 expression and clinical parameters maintained across all three datasets. These consistent patterns across independent cohorts suggest that our findings reflect genuine biological relationships rather than artifacts of sample size.
- Our findings align with existing literature regarding the roles of KPNA2 and FOXM1 in cell cycle regulation and cancer progression, providing biological plausibility to our observations. The mechanistic connection through the G2/M checkpoint pathway, revealed by GSEA, further supports the biological relevance of our findings.
- While we acknowledge that larger cohorts could provide additional statistical power, the multi-cohort approach we employed helps mitigate sample size limitations of any single dataset. This approach is widely accepted in biomarker research to establish reliability and generalizability of findings.
- We highlight the prognostic value of FOXM1 in the setting of KPNA2 high expression HR+HER2-BC patients and specify the potential therapeutic strategies targeting FOXM1 based on KPNA2 expression in HR+HER2-BC patients.
We welcome the opportunity to further validate these findings in larger cohorts, but believe the current evidence, supported by multiple independent datasets and mechanistic insights, provides a solid foundation for our conclusions."
We really appreciate your understanding and we have put your comment in the limitation of our results. Thank you very much.
Round 3
Reviewer 1 Report
Comments and Suggestions for Authors
I accept the answers